# Effect of Dynamic Preheating on the Thermal Behavior and Mechanical Properties of Laser-Welded Joints

**DOI:** 10.3390/ma15176159

**Published:** 2022-09-05

**Authors:** Linyi Xie, Wenqing Shi, Teng Wu, Meimei Gong, Detao Cai, Shanguo Han, Kuanfang He

**Affiliations:** 1School of Electronic and Information Engineering, Guangdong Ocean University, Zhanjiang 524088, China; 2Guangdong Provincial Key Laboratory of Advanced Welding Technology, China-Ukraine Institute of Welding, Guangdong Academy of Sciences, Guangzhou 510650, China; 3School of Mechatronic Engineering and Automation, Foshan University, Foshan 528000, China

**Keywords:** hybrid laser arc welding, laser welding, dynamic preheating, numerical simulation, yield strength

## Abstract

The high cooling rate and temperature gradient caused by the rapid heating and cooling characteristics of laser welding (LW) leads to excessive thermal stress and even cracks in welded joints. In order to solve these problems, a dynamic preheating method that uses hybrid laser arc welding to add an auxiliary heat source (arc) to LW was proposed. The finite element model was deployed to investigate the effect of dynamic preheating on the thermal behavior of LW. The accuracy of the heat transfer model was verified experimentally. Hardness and tensile testing of the welded joint were conducted. The results show that using the appropriate current leads to a significantly reduced cooling rate and temperature gradient, which are conducive to improving the hardness and mechanical properties of welded joints. The yield strength of welded joints with a 20 A current for dynamic preheating is increased from 477.0 to 564.3 MPa compared with that of LW. Therefore, the use of dynamic preheating to reduce the temperature gradient is helpful in reducing thermal stress and improving the tensile properties of the joint. These results can provide new ideas for welding processes.

## 1. Introduction

316L stainless steel is widely used in industry due to its excellent machinability and corrosion resistance [1,2]. Laser welding is usually used for 316L sheet welding. Laser welding (LW) entails using a laser heat source to melt material at a welded joint to form a molten pool and create a welded joint with excellent performance after solidification. This technique has broad application prospects in the welding process, as the high energy density and heat concentration of the laser beam leads to a small heat-affected zone and small deformation of the workpiece [3,4]. Therefore, LW is often used in sheet welding. However, the high energy density leads to rapid cooling and heating around the welded joint, which adversely affects the workpiece [5,6]. For example, the residual stress increases, the porosity of the welded joint increases, and the mechanical properties decrease. Alleviating the impact of these temperature effects is attracting increasing attention.

Asirvatham et al. [7] realized an accurate spatial distribution of laser energy by oscillating the laser beam, alleviating the high energy density of conventional LW and thus enabling better control of the geometric shape and microstructure of welded joints. Li et al. [8] used hollow beams to reduce the high energy density, which can effectively inhibit splash and porosity.

Preheating can reduce the temperature gradient between the joint and the substrate in the LW process, and it has an important influence on the thermal behavior, such as the temperature distribution and cooling rate in the welding process. Xiong et al. [9] found that preheating can reduce the thermal stress and cracking tendency in gas metal arc welding–based additive manufacturing. The preheating of the matrix is generally performed by heat transfer through contact with the heating table. However, this method has the disadvantages of large thermal damage, low efficiency, and low accuracy [10]. Therefore, some researchers proposed a dynamic preheating method [11]. Liu et al. [12] used a laser beam as a heat source to preheat a local area before moving the molten pool. They found that dynamic preheating helps to reduce thermal stress and decrease the probability of cracking. Shen et al. [13] found that the tensile strength and elongation of double-beam LW were improved compared with single-beam LW.

Hybrid laser arc welding (HLAW) is an efficient and deep penetration welding method, and a considerable amount of research was conducted [14,15,16]. Interestingly, the characteristics of lower arc pressure and low energy density are very suitable for dynamic preheating as an auxiliary heat source. HLAW is used to alleviate the rapid cooling and heat effect caused by LW and improve the mechanical properties of welded joints.

In this study, the influence of arc dynamic preheating in LW was discussed. A dynamic preheating method that uses HLAW to add an auxiliary heat source to LW was proposed. Experiment and numerical simulation were combined in the study of the welding process. The numerical simulation results were verified by using thermal imaging. The changes in the temperature distribution, temperature gradient, cooling rate, welding joint quality, and mechanical properties under different arc currents were analyzed.

## 2. Experimental Procedure

### 2.1. Experimental Methods and Equipment

In this study, the welding experiment was completed on a self-built HLAW system. As shown in Figure 1, the system consisted of a 10 kW disk laser (Trumpf TruDisk 10002) and a self-built arc control cabinet (PLAZER MP-1001-50). Laser characteristics are as follows in Table 1. The motion of the HLAW gun head was controlled by the arm of the six-axis Kuka robot. In the internal structure of the composite head, the angle between the tungsten level and the laser beam is 30°. During the experiment, the laser beam was vertically irradiated on the surface of a 316L steel plate. The arc temperature distribution and sample point temperature were collected through an infrared thermal imager (FLIR T640).

The butt-welding method was adopted in the welding experiment. The workpiece was a 100 mm × 50 mm × 1 mm 316L steel plate. Table 2 lists the mass component contents of 316L [17,18]. According to related research [19,20], the laser power is 600 W, the arc voltage is 20–23 V, the welding speed is 0.02 mm/s, and the nozzle height is 3 mm. The experimental welding parameters are given in Table 3. The protective gas used was pure argon, and the flow rate was 10 L/min.

After the welding experiment, the weld samples were processed, and the metallographic samples were prepared by mosaicing, grinding, and polishing. The microstructure was characterized using an optical microscope, and the microhardness of the welded joint was measured with a Wilson Vickers microhardness tester (Buehler VH1202). Vickers hardness analysis was performed for 10 s under a load of 0.2 kg. In order to evaluate the welded joints prepared by dynamic preheating, three standard tensile specimens were prepared for each experimental parameter. The detailed size of the tensile specimen is shown in Figure 2. The welded joint is perpendicular to the applied force in a tensile test. The tensile test was conducted on a universal tensile machine (MTS CMT5105) at a constant rate of 1 mm/min. The tensile fracture of welded joints was observed using a scanning electron microscope (FEI Quanta 250).

### 2.2. Numerical Simulation

The finite element model (FEM) was established using Ansys software. The overall framework of the numerical simulation is shown in Figure 3 [21,22,23]. The inverse bremsstrahlung of the laser and arc at 1030 nm can be ignored [3,24]. Therefore, two independent heat sources (laser and arc) were combined to form a composite heat source as the HLAW heat source in the experiment.

An appropriate heat source model is crucial to the accuracy and applicability of the simulation results. Given the large arc heat-affected zone, a new double-cone combined heat source was proposed, as shown in Figure 4. In LW, a single-cone heat source model was used, as shown in Figure 4a. In HLAW, a two-cone combined heat source model was used, as shown in Figure 4b. The Gaussian volume heat source model applied in this study exhibits linear attenuation in the negative-*Z*-axis direction [18,25]:(1)$$q=\frac{2\eta_{1}Pe}{\pi r_{1}{}^{2}d_{1}}e^{\left[-\frac{\left(x_{1}{}^{2}+y_{1}{}^{2}\right)}{r_{1}{}^{2}}\right]}\frac{z_{1}}{d_{1}}+\frac{2\eta_{2}UIe}{\pi r_{2}{}^{2}d_{2}}e^{\left[-\frac{\left(x_{2}{}^{2}+y_{2}{}^{2}\right)}{r_{2}{}^{2}}\right]}\frac{z_{2}}{d_{2}}$$
where *q* is the heat input, *η* is the heat source efficiency, *P* is the laser power, *r* is the heat source radius, *d* is the total depth of the heat source, *U* is the arc voltage, and *I* is the arc current. The subscripts 1 and 2 correspond to the laser and the arc, respectively.

In this experiment, only solid–thermal coupling was accounted for in the numerical simulation. Convection and radiation heat dissipation occurs on the surface around the substrate, and the composite convection coefficient *Q*_1_ (in W·m^−2^·K^−1^) was given by
(2)$$Q_{1}=\left\{\begin{array}{c} 0.0668T,\;T<773\;K,\\ 0.231T-82.1,\;T\geq 773\;K, \end{array}\right.$$
where *T* is temperature [18,26]. The position of sample point (point A) is shown in Figure 5. In the actual experiment, an infrared thermal imager was used to calibrate the numerical calculation and analysis, especially for the adjusted heat transfer coefficient, and the solid heat transfer coefficient of the material to the environment was 15 W·m^−2^·K^−1^ [27]. Given the large temperature gradient in the welding area, a nonlinear model was used for meshing, as shown in Figure 5 [28]. The minimum mesh side length size was 0.25 mm. There were 152,000 meshes in total and 792,568 nodes. The time step was set to 25 ms, open time integration was used, and workpiece cooling was set to 60 s.

The thermophysical parameters used in the calculations are summarized in Table 4 and Table 5 [17,29,30,31,32]. In order to simulate heat transfer in the molten pool, the thermal conductivity was set to triple that of room temperature when the temperature exceeded the melting point [33].

## 3. Results and Discussion

### 3.1. Temperature Test

The arc temperature distribution was collected using an infrared thermal imager, as shown in Figure 6. As can be seen from Figure 6a, the temperature of the central arc area with a current of 20 A was only 541 K. With a 40 A current, the temperature in the center of the arc reached 912 K, as can be seen from Figure 6b. This demonstrates the obvious influence of arc current on the preheating temperature. Moreover, the arc temperature distribution was symmetric about the *X* axis. This shows that the arc generated by the bias tungsten electrode passes through the nozzle and forms an approximate symmetrical arc, which supports the rationality of establishing the heat source model.

Figure 7a shows the temperature variation results obtained from the measurements and numerical simulation of temperature at the sampling points collected by the infrared thermal imager. The sample point coordinates are (30, 30, 1) (in units of millimeters). It can be clearly seen that the calculated temperature curve is in good agreement with the measured temperature curve. When the laser beam passes through the test point, the temperature reaches a peak value, and then it begins to decrease as a result of external cooling. The calculated peak temperature is slightly higher than the experimental value. This deviation can be attributed to the assumptions of the numerical model. Figure 7b,c show a comparison of the weld pool interface and metallographic weld pool section obtained by numerical simulation with currents of 0 and 20 A, respectively. It can be clearly seen that the calculated molten pool area is in good agreement with the measured molten pool section. From Figure 7a–c, it can be concluded that the established numerical model can well predict the thermal behavior of the welding process.

### 3.2. Temperature Distribution

The temperature distribution and shape of welds formed under LW and HLAW are shown in Figure 8a,b. By comparing Figure 8a,b, one can see that the high-temperature area around the molten pool in HLAW expands, and the heating area increases significantly, which is in line with the actual situation. HLAW entails greater heat input than LW, so the high-temperature zone around the molten pool expands. One can obviously see in Figure 7b,c that the pool area of HLAW is greater than that of LW, and the area of each temperature interval is also increased. Lei et al. [34] found that HLAW can effectively increase the molten pool area.

Figure 8c,d show local magnifications of Figure 8a,b, respectively. LW has an extremely high energy density and provides rapid cooling and heating in welded joints and heat-affected zones, as shown in Figure 8c. In the same temperature difference, HLAW has a larger buffer area than LW, which shows that HLAW can reduce the temperature gradient. Using arc dynamic preheating can therefore mitigate the rate of temperature change at welded joints.

Arc dynamic preheating improves the temperature uniformity of welded joints. To further study the influence of dynamic preheating of the arc current on the temperature gradient of the matrix, the temperature gradient and cooling rate change at multiple characteristic points (*x*, 0, 1) of each sample were obtained. Figure 9a shows the temperature gradient variation at a characteristic point (50, 0, 1) under different currents. The temperature gradient of LW was 425.0 K/mm, which was reduced to 244.8 K/mm after arc dynamic preheating. One can see that the temperature gradient of HLAW is significantly lower than that of LW. With the increase in current in HLAW, the temperature gradient increases and then decreases. The analysis indicates that this phenomenon is related to the preheating temperature, and the preheating temperature from different arc currents is different. Thus, with the increase in current, both the preheating range and the preheating temperature increase [35,36]. It is worth noting that the thermophysical properties and boundary conditions of the material are related to temperature. Therefore, different arc currents eventually lead to an increase in the temperature gradient and then a decrease. However, when the current is 40 A, the temperature gradient is reduced significantly to 171.9 K/mm, which is due to the increase in preheating temperature caused by excessive current, and the proportion of the laser heat input in the total heat input is reduced.

Figure 9b shows the change in cooling rate at a characteristic point (50, 0, 1) under different currents. The cooling rate of LW is 6676.0 K/s; this decreases to 2756.0 K/s after arc dynamic preheating. One can see that the cooling rate of HLAW is significantly lower than that of LW. In HLAW, the cooling rate increases with increasing current. The analysis indicates that this phenomenon is related to the actual temperature. The main factor affecting the cooling rate is the convective coefficient of the boundary conditions. With the increase in current, the heat input increases, and the corresponding actual temperature increases, resulting in an increase in the convective coefficient, so the cooling rate increases.

Figure 9c,d show the temperature gradient and cooling rate changes at multiple characteristic points (*x*, 0, 1) under different currents. One can see that the temperature gradient and cooling rate at each point along the *X* axis at the welding joint (*x*, 0, 1) are similar. Therefore, the characteristic point (50, 0, 1) can reflect the change in temperature gradient and cooling rate of the whole welded joint along the *X*-axis direction.

### 3.3. Hardness and Mechanical Properties

In order to confirm the effect of current on the properties of welded joints, the hardness of the middle section of welded joints was tested. Each sample had 15 indentation points, separated by 1 mm along the *Y* axis, as shown in Figure 10a. Figure 10b shows the microhardness curves for each sample. The hardness of the 316L base material (BM) was 185 ± 5 HV0.2. Notably, as the current increases, the hardness of the weld metal (WM) increases to 203 ± 2 HV0.2 when the current reaches 20 A. The hardness of WM under HLAW was higher than that under LW, but the hardness difference of WM under HLAW is not obvious under different currents.

The microstructure for the WMs is shown in Figure 11. Figure 11a–e show the cross-section microstructure of welded joints under 0 A, 10 A, 20 A, 30 A, and 40 A current. In Figure 11, a large amount of ferrite appears at the welded joint. By comparing Figure 11a and Figure 11b–e, it was found that the ferrite is refined after dynamic preheating. HLAW has an increased heat input compared to LW, which can be reflected in Figure 7 and Figure 8. However, in the same case, increasing the heat input coarsens the crystal structure [37]. This shows that the dynamic preheating process is the main impact on microstructure changes. According to Hall–Petch equation [38], grain refinement helps to improve microhardness. Interestingly, feathery ferrite was formed in Figure 11c–e. Chen et al. [18]. found that the precipitation of feathery ferrite would increase the local hardness of the weld. The formation of feathery ferrite is related to the cooling rate and temperature, which also confirms to some extent that feathery ferrite only exists in Figure 11c–e [18,39,40]. Therefore, dynamic preheating helps to increase the microhardness, which is also shown in Figure 10.

The tensile test results are shown in Figure 12. The yield strengths of specimens under 0 A, 10 A, 20 A, 30 A, and 40 A currents reached 477.0 MPa, 527.3 MPa, 564.3 MPa, 541.0 MPa, and 517.7 MPa, respectively. Thus, with the increase in current, the yield strength of the sample increases first and then decreases, reaching a maximum when the current is 20 A. However, the elongations under 0 A, 10 A, 20 A, 30 A, and 40 A currents were 35.3%, 36%, 42.6%, 37.2%, and 30.0%, respectively. Interestingly, the variation trend of elongation with current is similar to that of yield strength, revealing the obvious effect of arc dynamic preheating on tensile strength and elongation. That is, arc dynamic preheating helps to improve the tensile properties of welded joints. In Figure 11, the ferrite was refined after dynamic preheating, which helps increase the yield strength of welded joints. A smaller temperature gradient and cooling rate can significantly reduce the thermal stress of welded joints, thereby increasing the strength of welded joints. In Figure 9a,b, the temperature gradient and cooling rate were significantly reduced from LW to HLAW. These also confirm the results of the tensile test. In Figure 9a, the temperature gradient of 30 A and 40 A currents decreased significantly from 232.0 K/mm to 171.9 K/mm. It can be seen that the preheating temperature is obviously too large. Therefore, this is because the current is too large away from the optimal preheating temperature, leading to yield strength decreased at the current 40 A.

Figure 13a,b show the low magnification fracture microstructure of the sample joint at currents of 20 A and 40 A. At 20 A, the fracture roughness is low, the surface fluctuation is small, and dense dimples appear, the latter of which enhances the strength of the material. The fracture under a current of 40 A is mainly a tearing ridge. Figure 13c,d show enlarged micrographs of the central regions in Figure 13a,b, respectively. In Figure 13c, the region is composed of dense, deep dimples. This indicates that sufficient plastic deformation occurs in this region. This is related to the good uniformity of the sample preheated at an appropriate temperature, which contributes to the strength of the material. One can see from Figure 13d that the fracture surface is mainly a tear ridge and a shallow dimple, which indicates the low level of plastic deformation in this region. The above analysis is confirmed in Figure 12 by the tensile strength and elongation of welded joints.

## 4. Conclusions

(1)The FEM of dynamic preheating LW was successfully established, and the accuracy of the model was verified by experiments. The simulation results were in good agreement with the experimental results. Under the action of arc dynamic preheating, the molten pool area increases significantly;(2)Under the action of arc dynamic preheating, the temperature gradient and cooling rate of HLAW were significantly lower than those of LW. Changing the current level has a certain influence on the temperature gradient. Excessive current (40 A) leads to a significant decrease in the temperature gradient of the preheating temperature rise. However, the change in current has a negligible effect on the cooling rate;(3)Arc dynamic preheating is conducive to improving the hardness and tensile properties of welded joints due to ferrite refinement after dynamic preheating. Feathery ferrite forms at appropriate preheating temperature. Compared with values from LW, the yield strength of welded joints with dynamic preheating by a current of 20 A increased by 18.3%, from 477.0 to 564.3 MPa. The use of the appropriate current is helpful in reducing thermal stress and improving the tensile properties of the joint.

## Figures and Tables

**Figure 1 materials-15-06159-f001:**
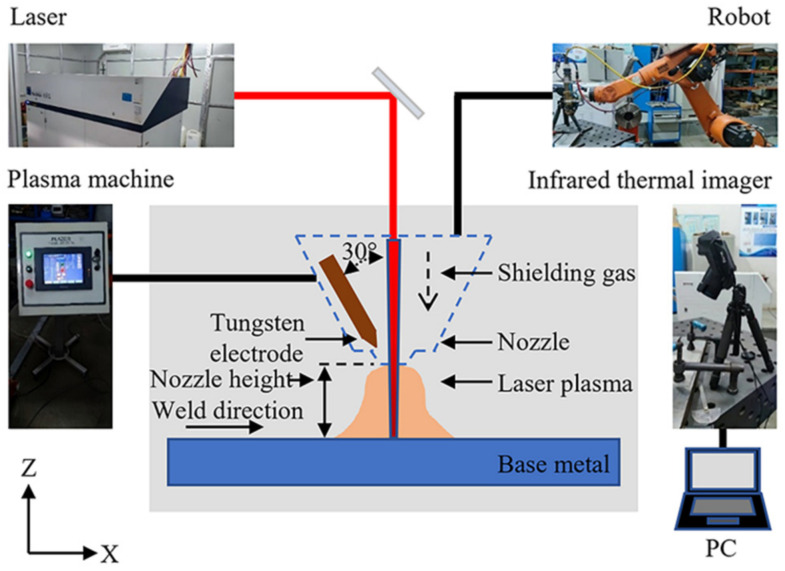
HLAW system.

**Figure 2 materials-15-06159-f002:**
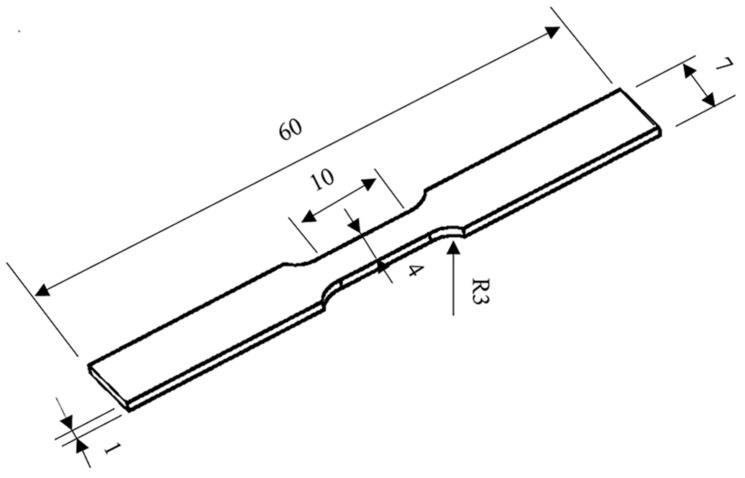
Tensile sample showing dimensions (in millimeters).

**Figure 3 materials-15-06159-f003:**
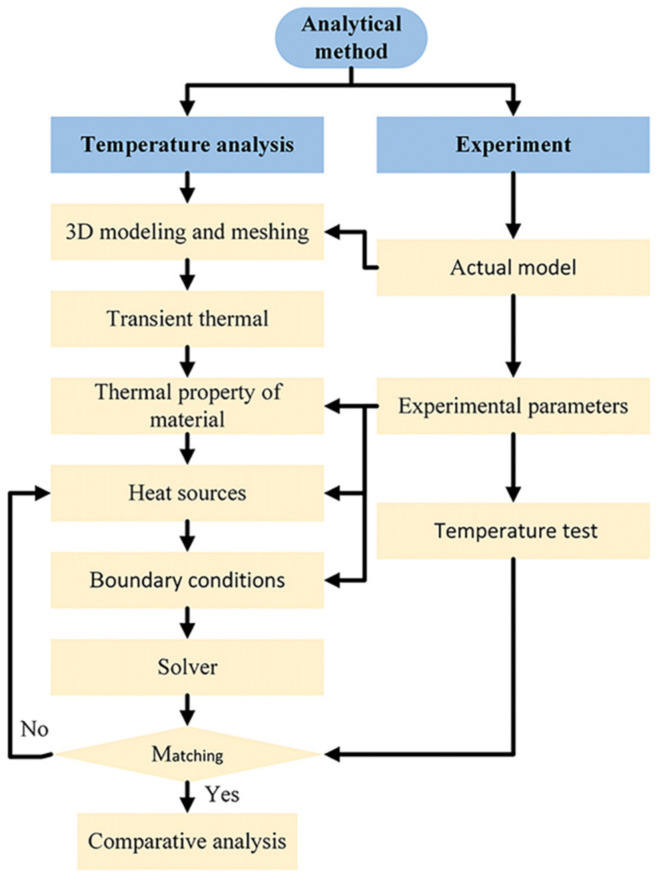
FEM simulation framework.

**Figure 4 materials-15-06159-f004:**
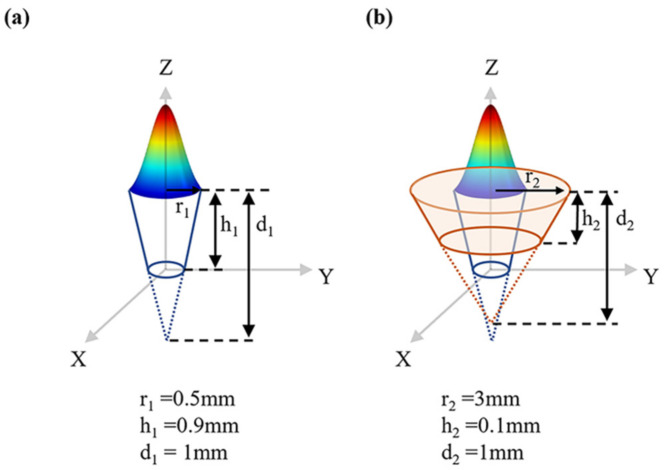
Gauss heat source model: (**a**) LW; (**b**) HLAW.

**Figure 5 materials-15-06159-f005:**
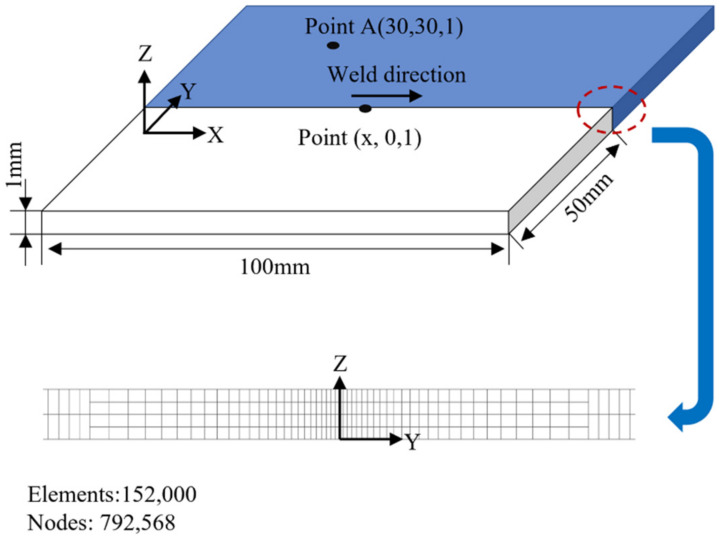
Mesh of FEM.

**Figure 6 materials-15-06159-f006:**
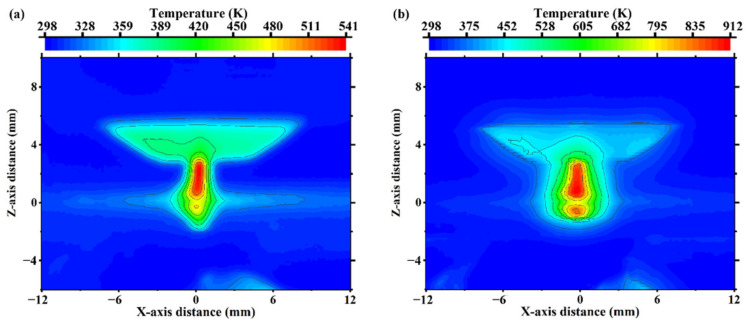
Temperature distribution of the arc with currents of (**a**) 20 A and (**b**) 40 A.

**Figure 7 materials-15-06159-f007:**
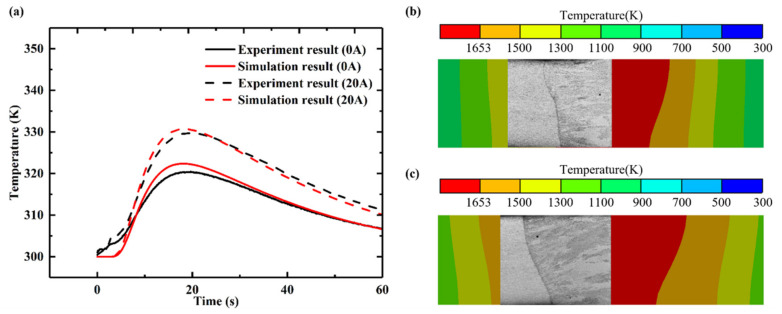
Comparison of experimental temperature verification. (**a**) Temperature curve at point A (30,30,1). (**b**) Comparison of cross-section molten pool morphology with a 0 A current. (**c**) Comparison of cross-section molten pool morphology with a 20 A current.

**Figure 8 materials-15-06159-f008:**
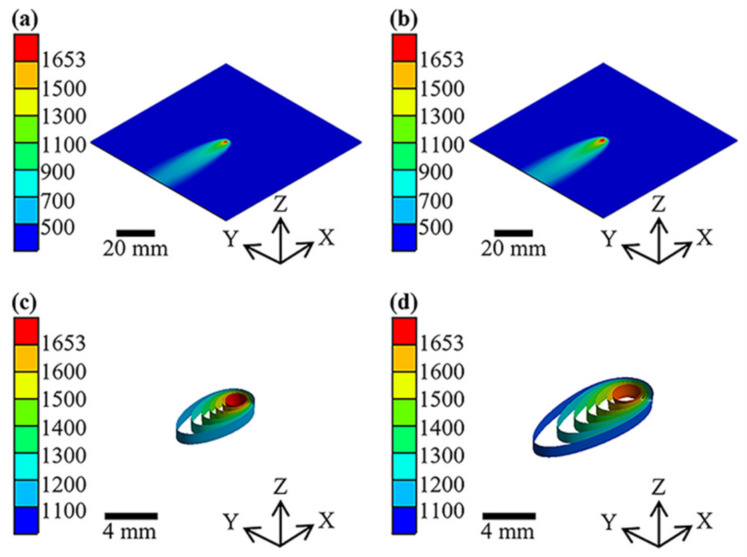
Temperature distribution at 2.5 s with currents of (**a**) 0 A and (**b**) 20 A. (**c**,**d**) Magnified views of the central region taken from the central areas in (**a**,**b**), respectively.

**Figure 9 materials-15-06159-f009:**
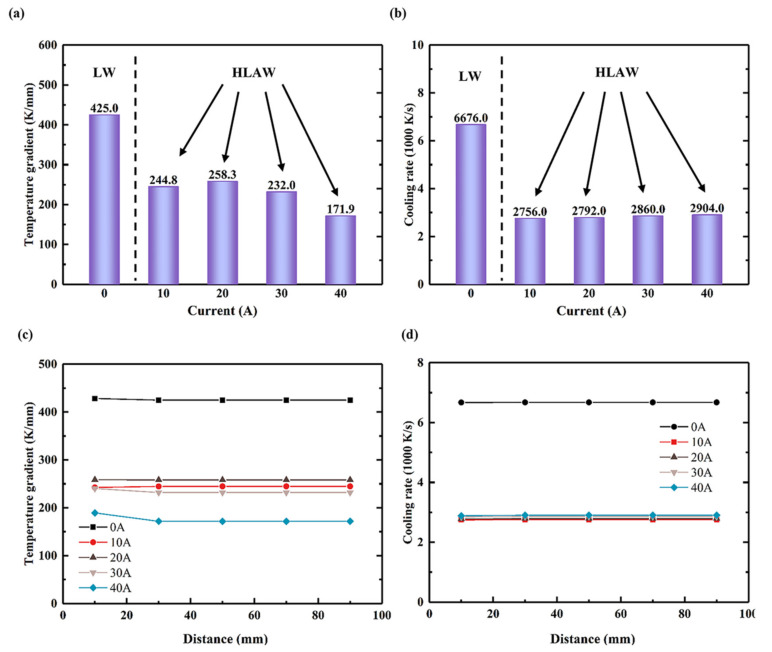
Temperature distribution with different currents at special points. (**a**) Maximum temperature gradients at special points (50, 0, 1). (**b**) Maximum cooling rate at special points (50, 0, 1). (**c**) Maximum temperature gradients at special points (*x*, 0, 1). (**d**) Maximum cooling rate at special points (*x*, 0, 1).

**Figure 10 materials-15-06159-f010:**
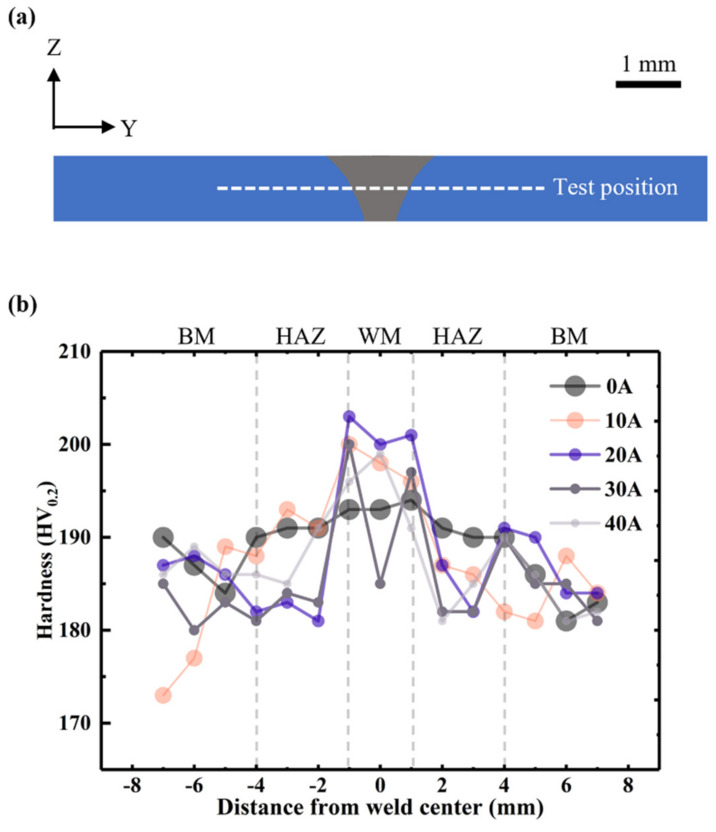
Microhardness distribution in the weld section. (**a**) Microhardness test position. (**b**) Microhardness test results.

**Figure 11 materials-15-06159-f011:**
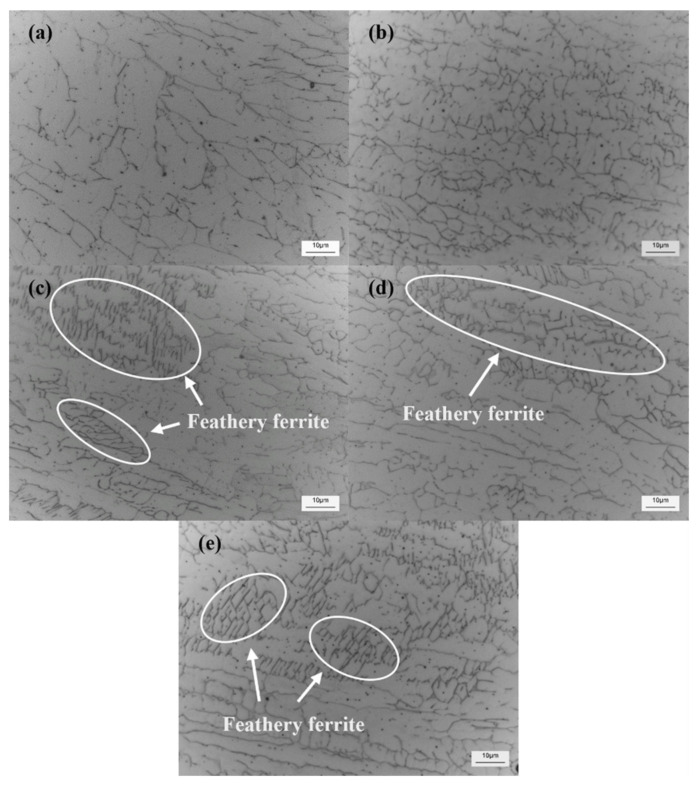
Microstructural images obtained from the optical microscope for the WMs. (**a**–**e**) 0 A, 10 A, 20 A, 30 A, and 40 A currents, respectively.

**Figure 12 materials-15-06159-f012:**
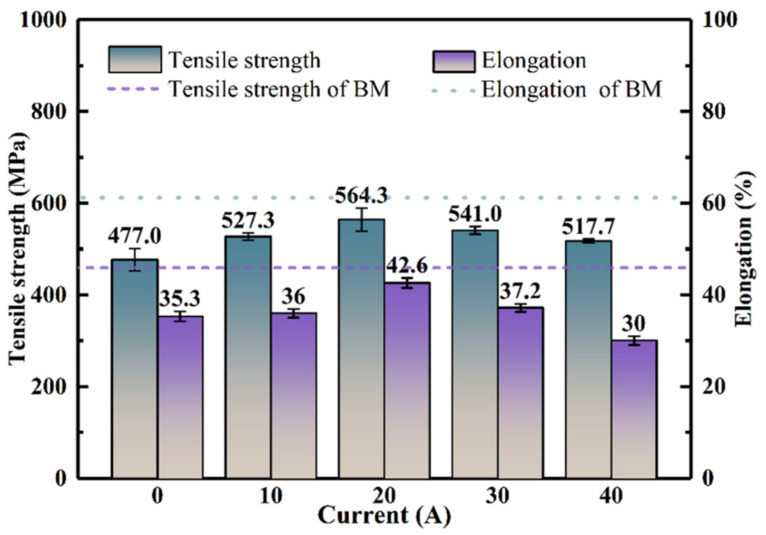
Tensile mechanical properties.

**Figure 13 materials-15-06159-f013:**
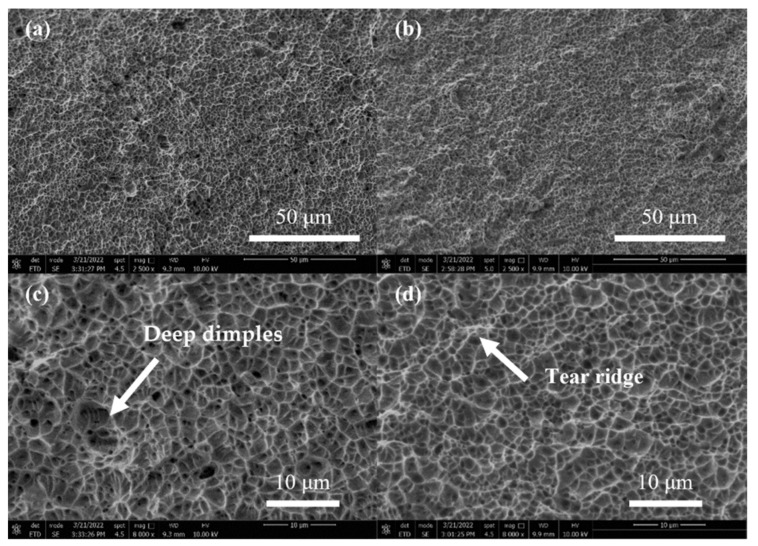
Scanning electron microscope images of the fracture for (**a**) 20 A and (**b**) 40 A cases. (**c**) and (**d**) Higher magnification micrographs taken from the central areas in (**a**) and (**b**), respectively.

**Table 1 materials-15-06159-t001:** Laser characteristics.

Laser Characteristic Parameters	Value
maximum continuous output power	10,000 W
Power output stability	±1%
beam quality	8 mm·mrad
laser wavelength	1030 nm
Fiber diameter	400 μm

**Table 2 materials-15-06159-t002:** Mass fraction of 316L (wt %).

Type	Fe	Si (%)	Mo (%)	Cr (%)	Ni (%)	Mn (%)
316L	Balance	≤1	2–3	16–18	12–15	≤2

**Table 3 materials-15-06159-t003:** Test scheme.

Case	1	2	3	4	5
Current (A)	0	10	20	30	40

**Table 4 materials-15-06159-t004:** 316L thermophysical parameters.

Temperature (K)	Density (kg·m^−3^)	Specific Heat (J·kg^−1^·K^−1^)	Thermal Conductivity (W·m^−1^·K^−1^)
300	7954	498.73	13.44
500	7864	525.51	16.8
700	7771	551.87	19.87
900	7674	578.65	22.79
1100	7574	605.01	25.46
1300	7471	631.78	28.02

**Table 5 materials-15-06159-t005:** Thermophysical parameters assumed in the computer simulations.

Nomenclature	Value
Ambient temperature	300 K
Solidus temperature	1653 K
Liquidus temperature	1731 K
Gasification temperature	3134 K
Latent heat of fusion	2.77 × 10^5^ K/kg
Latent heat of evaporation	6.34 × 10^6^ K/kg

## Data Availability

Not applicable.

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
