# Peer review of "Effect of Dynamic Preheating on the Thermal Behavior and Mechanical Properties of Laser-Welded Joints"

_materials, 2022, doi:10.3390/ma15176159_

Round 1
Reviewer 1 Report
1. The results are good but the weakness point is the discussion of the results. The presented results need to be discussed deeply as well as linking the achieved results with the literature.
2. What are the properties of the used laser? please add a table of them.
3. Why did you use 600 W and 0.02 mm/s ?
4. How did you measure the mass fraction of 316L listed in table 1? if they are standard please add a reference?
5. In equation 2, for what 773 refers to?
Reviewer 2 Report
Effect of dynamic preheating on the thermal behavior and mechanical properties of laser-welded joints
1) Please add the novelty of the work at the end of Introduction section.
2) Table 1 shows the standard mass fraction of 316L (wt %), used in the study. But steel used here has definite composition. Why range of composition is given in the table. Please mention the composition of 316L used in this study.
3) Also mention the method used to determine the chemical composition.
4) In figure 2, please mention the standard for tensile specimen.
5) Please mention the literature support for Table (3 and 4) details.
6) Please mention the assumptions used for the Numerical Simulations for heating and cooling cycles carried out.
7) Below Figure 8 title, in the text, it is mentioned as “Arc dynamic preheating improves the temperature uniformity of welded joints” Please give the literature support.
8) Detailed justification is required for the reduction in the yield strength of specimen with the increase in applied current. Please add to the manuscript.
9) In Figure 12, please incorporate the details for self-explanation.
10) Effort is required to correlate the microstructure and tensile properties. Please improve the quality of the manuscript by comparing the welding parameters, microstructure and tensile properties.
11) Please add scope for further research to minimise the short comings.
12) In REFERENCE, the page numbers from - to be given like, 227 to 235, whereas some references only beginning of the page, like 225 is mentioned. Please maintain the format uniformity.
Author Response
Dear Editors and Reviewers:
Thank you so much for giving us an opportunity to revise our manuscript entitled “Effect of dynamic preheating on the thermal behavior and mechanical properties of laser-welded joints” (Manuscript ID: materials-1872595). We highly appreciate the reviewers’ and editors’ constructive comments and suggestions on our manuscript, which is really helpful for us. We have studied the comments very carefully and have accordingly made revisions marked in yellow in the revised version, which we would like to submit for your kind consideration. As attached, please also find the detailed reply to the reviewers’ comments. We hope that the revision and correction will meet with approval.
- Please add the novelty of the work at the end of Introduction section.
Response: Considering the Reviewer’s suggestion, we add the novelty of the work at the end of Introduction section. A dynamic preheating method that uses HLAW to add an auxiliary heat source to LW is proposed. Experiment and numerical simulation are combined in the study of welding process.
- Table 1 shows the standard mass fraction of 316L (wt %), used in the study. But steel used here has definite composition. Why range of composition is given in the table. Please mention the composition of 316L used in this study.
Response: We are very sorry for our unclear expression. The standard mass fraction of 316L refers to the references[1,2]. We added the references and listed below.
- Also mention the method used to determine the chemical composition.
Response: We are very sorry for our negligence. The standard mass fraction of 316L refers to the references[1,2].
- In figure 2, please mention the standard for tensile specimen.
Response: This is a non-standard stretch size. We give the following reasons why choose this size. Sample size cannot be cut to meet ASTM E8 American Standard. Referring to the relevant literature[3,4], we have scaled to obtain similar tensile dimensions. At the same time, we conducted a tensile test on the raw materials, the results as an evaluation standard with the standard test in Figure 12.
- Please mention the literature support for Table (3 and 4) details.
Response: We are very sorry for our unclear expression. The thermophysical parameters used in the calculations are summarized in Tables 4 and 5 (Tables 3 and 4 in Unmodified manuscripts)[1,5-8].
- Please mention the assumptions used for the Numerical Simulations for heating and cooling cycles carried out.
Response: We are very sorry for our unclear expression. In this experiment, only solid–thermal coupling was accounted for in the numerical simulation. Convection and radiation heat dissipation occur on the surface around the substrate and the composite convection coefficient Q1 (in W·m−2·K−1) was given by
where T is temperature[2,9]. In the actual experiment, an infrared thermal imager was used to calibrate the numerical calculation and analysis, especially for the adjusted heat transfer coefficient, and the solid heat transfer coefficient of the material to the environment was 15 W·m−2·K−1[10]. To simulate heat transfer in the molten pool, the thermal conductivity was set to triple that of room temperature when the temperature exceeded the melting point.
- Below Figure 8 title, in the text, it is mentioned as “Arc dynamic preheating improves the temperature uniformity of welded joints” Please give the literature support.
Response: We are very sorry for our unclear expression. Above Figure 8 title, “In the same temperature difference, HLAW has a larger buffer area than LW, which show that HLAW can reduce the temperature gradient. Using arc dynamic preheating can therefore mitigate the rate of temperature change at welded joints.” The temperature difference in the same area interval is reduced, that is, the temperature change decreases. So say “Arc dynamic preheating improves the temperature uniformity of welded joints”.
- Detailed justification is required for the reduction in the yield strength of specimen with the increase in applied current. Please add to the manuscript.
Response: We are very sorry for our negligence. We believe that this is due to the excessive current away from the optimal preheating temperature resulting in a decrease in yield strength. In Figure 9a, the temperature gradient of 30A and 40A currents decreased significantly from 232.0 K / mm to 171.9 K / mm. It can be seen that the preheating temperature is obviously too large.
- In Figure 12, please incorporate the details for self-explanation.
Response: Considering the Reviewer’s suggestion. We add details in Section 3.3. In Figure 13c, the region is composed of dense, deep dimples. This indicates that sufficient plastic deformation occurs in this region. This is related to the good uniformity of the sample preheated at an appropriate temperature, which contributes to the strength of the material. One can see from Figure 13d that the fracture surface is mainly a tear ridge and a shallow dimple, which indicates the low level of plastic deformation in this region. We believe that this is due to the excessive current away from the optimal preheating temperature resulting in a decrease in yield strength. In Figure 9a, the temperature gradient of 30A and 40A currents decreased significantly from 232.0 K / mm to 171.9 K / mm. It can be seen that the preheating temperature is obviously too large. The above analysis is confirmed in Figure 12 by the tensile strength and elongation of welded joints.
- Effort is required to correlate the microstructure and tensile properties. Please improve the quality of the manuscript by comparing the welding parameters, microstructure and tensile properties.
Response: Considering the Reviewer’s suggestion. We add details in Section 3.3. Microstructure for the WMs is shown in Figure 11. Figures.11a-e are the cross-section microstructure of welded joints under 0A, 10A, 20A, 30A and 40A current. In Figure 11, a large amount of ferrite appears at the welded joint. Comparing Figure 11a and Figures 11b-e, it is found that the ferrite is refined after dynamic preheating. HLAW is increased heat input than LW, which can be reflected in Figure 7 and Figure 8. However, in the same case, increasing the heat input coarsens the crystal structure[11]. This shows that the dynamic preheating process is the main impact on microstructure changes. According to Hall-Petch equation[12], grain refinement helps to improve microhardness. Interestingly, feathery ferritewas formed in Figures.11c-e. Chen et al.[2] found that the precipitation of feathery ferrite would increase the local hardness of the weld. The formation of feathery ferrite is related to the cooling rate and temperature, which also confirms to some extent that feathery ferrite only exists in Figures. 11c-e[2,13,14]. Therefore, dynamic preheating helps to increase the microhardness, which is also shown in Figure 10. In Figure 11, the ferrite is refined after dynamic preheating, which helps increase the yield strength of welded joints.
Figure 11. Microstructural images obtained from the optical microscope for the WMs. (a-e) 0A, 10A, 20A, 30A and 40A currents, respectively.
- Please add scope for further research to minimise the short comings.
Response: Considering the Reviewer’s suggestion. We add details about microstructure in Section 3.3.
- In REFERENCE, the page numbers from - to be given like, 227 to 235, whereas some references only beginning of the page, like 225 is mentioned. Please maintain the format uniformity.
Response: We are very sorry for our negligence. We modify as required in the references. The page number of some references is uncertain, so we added DOI.
We appreciate for Reviewers warm work earnestly, and hope that the correction will meet with approval. Once again, thank you very much for your comments and suggestions.
References:
- Gao, J.; Wu, C.; Hao, Y.; Xu, X.; Guo, L. Numerical Simulation and Experimental Investigation On Three-Dimensional Modelling of Single-Track Geometry and Temperature Evolution by Laser Cladding. Optics & Laser Technology 2020, 129, https://doi.org/10.1016/j.optlastec.2020.106287
- Chen, L.; Mi, G.; Zhang, X.; Wang, C. Numerical and Experimental Investigation On Microstructure and Residual Stress of Multi-Pass Hybrid Laser-Arc Welded 316L Steel. Materials & Design 2019, 168, https://doi.org/10.1016/j.matdes.2019.107653
- Tembhurkar, C.; Kataria, R.; Ambade, S.; Verma, J.; Sharma, A.; Sarkar, S. Effect of Fillers and Autogenous Welding On Dissimilar Welded 316L Austenitic and 430 Ferritic Stainless Steels. J. Mater. Eng. Perform. 2021, 30 (2), 1444-1453. https://doi.org/10.1007/s11665-020-05395-4
- Shen, J.; Li, B.; Hu, S.; Zhang, H.; Bu, X. Comparison of Single-Beam and Dual-Beam Laser Welding of Ti–22Al–25Nb/TA15 Dissimilar Titanium Alloys. Optics & Laser Technology 2017, 93, 118-126. https://doi.org/10.1016/j.optlastec.2017.02.013
- Maurin, A. Numerical Investigation of Degradation of 316L Steel Caused by Cavitation. Materials 2021, 14 (11), https://doi.org/10.3390/ma14113131
- Liu, L.; Huang, M.; Ma, Y.H.; Qin, M.L.; Liu, T.T. Simulation of Powder Packing and Thermo-Fluid Dynamic of 316L Stainless Steel by Selective Laser Melting. J. Mater. Eng. Perform. 2020, 29 (11), 7369-7381. https://doi.org/10.1007/s11665-020-05230-w
- Rubenchik, A.; Wu, S.; Mitchell, S.; Golosker, I.; LeBlanc, M.; Peterson, N. Direct Measurements of Temperature-Dependent Laser Absorptivity of Metal Powders. Appl Opt 2015, 54 (24), 7230-3. https://doi.org/10.1364/AO.54.007230
- Masmoudi, A.; Bolot, R.; Coddet, C. Investigation of the Laser–Powder–Atmosphere Interaction Zone During the Selective Laser Melting Process. Journal of Materials Processing Tech. 2015, 225, https://doi.org/10.1016/j.jmatprotec.2015.05.008
- Yang, Z.; Fang, Y.; He, J. Numerical Simulation of Heat Transfer and Fluid Flow During Vacuum Electron Beam Welding of 2219 Aluminium Girth Joints. Vacuum 2020, 175, https://doi.org/10.1016/j.vacuum.2020.109256
- Ahmad, S.N.; Manurung, Y.H.P.; Prajadhiana, K.P.; Busari, Y.O.; Mat, M.F.; Muhammad, N.; Leitner, M.; Saidin, S. Numerical Modelling and Experimental Analysis On Angular Strain Induced by Bead-On-Plate SS316L GMAW Using Inherent Strain and Thermomechanical Methods. The International Journal of Advanced Manufacturing Technology 2022, 120 (1-2), 627-644. https://doi.org/10.1007/s00170-022-08684-5
- Zhao, Y.; You, J.; Qin, J.; Dong, C.; Liu, L.; Liu, Z.; Miao, S. Stationary Shoulder Friction Stir Welding of Al–Cu Dissimilar Materials and its Mechanism for Improving the Microstructures and Mechanical Properties of Joint. Materials Science and Engineering: A 2022, 837, https://doi.org/10.1016/j.msea.2022.142754
- Hu, Y.; Zhao, Y.; Peng, Y.; Yang, W.; Ma, X.; Wang, B. High-Strength Joint of Nuclear-Grade FeCrAl Alloys Achieved by Friction Stir Welding and its Strengthening Mechanism. Journal of Manufacturing Processes 2021, 65, 1-11. https://doi.org/10.1016/j.jmapro.2021.03.007
- Vadiraj, A.; Balachandran, G.; Kamaraj, M.; Gopalakrishna, B.; Prabhakara Rao, K. Structure–Property Correlation in Austempered Alloyed Hypereutectic Gray Cast Irons. Materials Science and Engineering: A 2010, 527 (3), 782-788. https://doi.org/10.1016/j.msea.2009.08.074
- Banavasi Shashidhar, M.; Ravishankar, K.S.; Naik Padmayya, S. Influence of High Mn-Cu-Mo On Microstructure and Fatigue Characteristics of Austempered Ductile Iron. IOP conference series. Materials Science and Engineering 2018, 330 (1), https://doi.org/10.1088/1757-899X/330/1/012020
Reviewer 3 Report
In this paper, the influence of dynamic arc preheating on LW of one of the most popular stainless steels, AISI 316L, is discussed. In the introductory chapter, the authors analyzed knowledge about laser welding. However, they did not pay attention to the material used in terms of welding. It is requested to be supplemented from this area as well. In the methodology part of the experimental research, the used equipment and procedures during the experiment are described. It is necessary to supplement the result for the analysis of the microstructure, the direction of the weld joint with respect to the loading force during the tensile test (perpendicular or parallel, or according to the standard of the manufactured sample for the tensile test). Based on the experimental results, the authors state that the hardness of the base material 316L was 185 ± 5 HV0.2. It is not enough to say that it is remarkable that as the current increases, the hardness of the weld metal (WM) increases to 203 ± 2 HV0.2 when the current reached 20 A. It is necessary to supplement (explain) what this is due to (structure, microstructure, see Schaeffler Diagram, etc.). Figure 10, why is there less hardness in some samples in the HAZ area than in the BM? Figure 11 shows the tensile strength and in the text in lines 203 and 204 the authors state the yield strength "The yield strengths of specimens under 0-, 10-, 20-, 30- and 40-A currents reached 477.0, 527.3, 564.3, 541.0, and 517.7 MPa". What is correct? Referring to Figure 10 , for example at 20 A current the hardness in the WM region is over 200 HV0.2, in the HAZ region the lowest is about 180 MPa but the yield strengths are the highest at 541 MPa. If we assume that the welded joint is run perpendicular to the applied force, then deformation or failure of the specimen should occur at the point where the strength is lowest. Is my reasoning correct? Based on the results obtained, the conclusions are appropriately formulated. Perhaps a comment on the first conclusion. What do the authors consider to be a good agreement of the results?
Author Response
Dear Editors and Reviewers:
Thank you so much for giving us an opportunity to revise our manuscript entitled “Effect of dynamic preheating on the thermal behavior and mechanical properties of laser-welded joints” (Manuscript ID: materials-1872595). We highly appreciate the reviewers’ and editors’ constructive comments and suggestions on our manuscript, which is really helpful for us. We have studied the comments very carefully and have accordingly made revisions marked in yellow in the revised version, which we would like to submit for your kind consideration. As attached, please also find the detailed reply to the reviewers’ comments. We hope that the revision and correction will meet with approval.
- However, they did not pay attention to the material used in terms of welding. It is requested to be supplemented from this area as well.
Response: Considering the Reviewer’s suggestion, the materials used in welding shall be supplemented as required, and the corresponding modifications shall be made in the abstract. 316L stainless steel is widely used in industry due to its excellent machinability and corrosion resistance[1,2]. Laser welding is usually used for 316L sheet welding.
- It is necessary to supplement the result for the analysis of the microstructure, the direction of the weld joint with respect to the loading force during the tensile test (perpendicular or parallel, or according to the standard of the manufactured sample for the tensile test).
Response: Considering the Reviewer’s suggestion, the microstructure of the analysis results is supplemented as required, and the corresponding modifications are made in Section 3.3. In the tensile test, the welded joint is perpendicular to the applied force, which is not explained. It is supplemented in Section 2.1. In figure 11, a large amount of ferrite appears at the welded joint. Comparing Figure 11a and Figures 11b-e, it is found that the ferrite is refined after dynamic preheating. HLAW is increased heat input than LW, which can be reflected in Figure 7 and Figure 8. However, in the same case, increasing the heat input coarsens the crystal structure[3]. This shows that the dynamic preheating process is the main impact on microstructure changes. Interestingly, feathery ferritewas formed in Figures.11c-e. Chen et al[4]. found that the precipitation of feathery ferrite would increase the local hardness of the weld.
- Based on the experimental results, the authors state that the hardness of the base material 316L was 185 ± 5 HV0.2. It is not enough to say that it is remarkable that as the current increases, the hardness of the weld metal (WM) increases to 203 ± 2 HV0.2 when the current reached 20 A. It is necessary to supplement (explain) what this is due to (structure, microstructure, see Schaeffler Diagram, etc.).
Response: We are very sorry for our negligence. According to the requirements of supplementary analysis of the reasons for changes in hardness. In figure 11, a large amount of ferrite appears at the welded joint. Comparing Figure 11a and Figures 11b-e, it is found that the ferrite is refined after dynamic preheating. According to Hall-Petch equation[5], grain refinement helps to improve microhardness. Interestingly, feathery ferritewas formed in Figures.11c-e. Chen et al[4]. found that the precipitation of feathery ferrite would increase the local hardness of the weld. The formation of feathery ferrite is related to the cooling rate and temperature, which also confirms to some extent that feathery ferrite only exists in Figures. 11c-e[4,6,7]. Therefore, dynamic preheating helps to increase the microhardness, which is also shown in Figure 10.
Figure 11. Microstructural images obtained from the optical microscope for the WMs. (a-e) 0A, 10A, 20A, 30A and 40A currents, respectively.
- Figure 10, why is there less hardness in some samples in the HAZ area than in the BM?
Response: The hardness of heat affected zone of the sample is concentrated in 180-190 HV0.2, which is close to the hardness of 316L substrate which is 185±5 HV0.2. In Figure 10, the hardness of some samples in the heat affected zone is lower than that of BM, which is a slight phenomenon. Amborish et al.[8] also appeared this phenomenon in the hardness test.
- Figure 11 shows the tensile strength and in the text in lines 203 and 204 the authors state the yield strength "The yield strengths of specimens under 0-, 10-, 20-, 30- and 40-A currents reached 477.0, 527.3, 564.3, 541.0, and 517.7 MPa". What is correct? Referring to Figure 10, for example at 20 A current the hardness in the WM region is over 200 HV0.2, in the HAZ region the lowest is about 180 MPa but the yield strengths are the highest at 541 MPa. If we assume that the welded joint is run perpendicular to the applied force, then deformation or failure of the specimen should occur at the point where the strength is lowest. Is my reasoning correct? Based on the results obtained, the conclusions are appropriately formulated. Perhaps a comment on the first conclusion. What do the authors consider to be a good agreement of the results?
Response: We are very sorry for our unclear expression. The deformation or failure of the sample should occur at the point of lowest strength, but the hardness cannot completely represent the strength. In other words, a low hardness does not mean a low strength specimen. The strength of the specimen is visually represented by the internal stress and is related to the microstructure, welding defects, machining process, etc[3,9]. In this paper, the temperature gradient and cooling rate of welded joints of each sample are obtained by numerical simulation. Smaller temperature gradient and cooling rate can significantly reduce the thermal stress of welded joints, thereby increasing the strength of welded joints. This is in good agreement with the strength of the specimens obtained by tensile testing.
We appreciate for Reviewers warm work earnestly, and hope that the correction will meet with approval. Once again, thank you very much for your comments and suggestions.
References:
- Tembhurkar, C.; Kataria, R.; Ambade, S.; Verma, J.; Sharma, A.; Sarkar, S. Effect of Fillers and Autogenous Welding On Dissimilar Welded 316L Austenitic and 430 Ferritic Stainless Steels. J. Mater. Eng. Perform. 2021, 30 (2), 1444-1453. https://doi.org/10.1007/s11665-020-05395-4
- Thomas, M.; Prakash, R.V.; Ganesh Sundara Raman, S.; Vasudevan, M. High Temperature Fatigue Crack Growth Rate Studies in Stainless Steel 316L(N) Welds Processed by A-TIG and MP-TIG Welding. MATEC Web of Conferences 2018, 165, https://doi.org/10.1051/matecconf/201816521014
- Zhao, Y.; You, J.; Qin, J.; Dong, C.; Liu, L.; Liu, Z.; Miao, S. Stationary Shoulder Friction Stir Welding of Al–Cu Dissimilar Materials and its Mechanism for Improving the Microstructures and Mechanical Properties of Joint. Materials Science and Engineering: A 2022, 837, https://doi.org/10.1016/j.msea.2022.142754
- Chen, L.; Mi, G.; Zhang, X.; Wang, C. Numerical and Experimental Investigation On Microstructure and Residual Stress of Multi-Pass Hybrid Laser-Arc Welded 316L Steel. Materials & Design 2019, 168, https://doi.org/10.1016/j.matdes.2019.107653
- Hu, Y.; Zhao, Y.; Peng, Y.; Yang, W.; Ma, X.; Wang, B. High-Strength Joint of Nuclear-Grade FeCrAl Alloys Achieved by Friction Stir Welding and its Strengthening Mechanism. Journal of Manufacturing Processes 2021, 65, 1-11. https://doi.org/10.1016/j.jmapro.2021.03.007
- Vadiraj, A.; Balachandran, G.; Kamaraj, M.; Gopalakrishna, B.; Prabhakara Rao, K. Structure–Property Correlation in Austempered Alloyed Hypereutectic Gray Cast Irons. Materials Science and Engineering: A 2010, 527 (3), 782-788. https://doi.org/10.1016/j.msea.2009.08.074
- Banavasi Shashidhar, M.; Ravishankar, K.S.; Naik Padmayya, S. Influence of High Mn-Cu-Mo On Microstructure and Fatigue Characteristics of Austempered Ductile Iron. IOP conference series. Materials Science and Engineering 2018, 330 (1), https://doi.org/10.1088/1757-899X/330/1/012020
- Sriba, A.; Bouquerel, J.; Vogt, J. DIC-aided Analysis of the Fatigue Behaviour of a Welded 316L Stainless Steel. Weld. World 2022, 66 (9), 1915-1927. https://doi.org/10.1007/s40194-022-01355-9
- Hedhibi, A.C.; Touileb, K.; Djoudjou, R.; Ouis, A.; Alrobei, H.; Ahmed, M.M.Z. Mechanical Properties and Microstructure of TIG and ATIG Welded 316L Austenitic Stainless Steel with Multi-Components Flux Optimization Using Mixing Design Method and Particle Swarm Optimization (PSO). Materials 2021, 14 (23), https://doi.org/10.3390/ma14237139
Round 2
Reviewer 1 Report
The authors have addressed adequately most of the question raised in the previous review. The manuscript is ready for publication